# Exercise Interventions Improved Sleep Quality through Regulating Intestinal Microbiota Composition

**DOI:** 10.3390/ijerph191912385

**Published:** 2022-09-28

**Authors:** Liangwu Qiu, Fuhong Gong, Jiang Wu, Dingyun You, Yinzhou Zhao, Lianwu Xu, Xue Cao, Fukai Bao

**Affiliations:** 1Department of Physical Education, Kunming Medical University, Kunming 650500, China; 2Faculty of Basic Medical Science, Kunming Medical University, Kunming 650500, China; 3Biomedical Engineering Research Center, Yunnan Key Laboratory of Stem Cell and Regenerative Medicine, Kunming Medical University, Kunming 650500, China; 4Department of Laboratory Animal Science, Kunming Medical University, Kunming 650500, China

**Keywords:** exercise intervention, sleep quality, intestinal flora, microbiota composition

## Abstract

(1) Background: Sleep quality is closely related to the physical and mental health of college students. The objectives of this study were to obtain data on the sleep quality of university students and to investigate the relationship between intestinal flora and the improvement in sleep quality through exercise intervention. (2) Methods: Here, 11 university students with a body mass index (BMI) ≤ 18 and Pittsburgh Sleep Quality Index (PSQI) ≥ 7 were selected as experimental subjects, and another 11 healthy people were recruited as control subjects. The experimental group and control group were each intervened with exercise for 8 weeks. We used 16SrDNA sequencing technology to analyze the variations of the intestinal flora and the relation of the variations and sleep quality improvement between the experimental group and the control group before and after the exercise intervention. (3) Results: The differences in gut flora composition between people with sleep disorders and healthy people were statistically significant (*p* < 0.05). Before and after the exercise intervention, the differences were also statistically significant (*p* < 0.05) in people with sleep disorders. The sleep-disordered population had a larger proportion compared with the healthy population (*p* < 0.05). *Blautia* and *Eubacterium hallii* were microbe markers in the sleep-disordered population before and after the exercise intervention, while there was no microbe marker found in the healthy population. (4) Conclusions: The increase in *Blautia* and *Eubacterium hallii,* and the decrease in *Agathobacter* are associated with healthy sleep. Gut flora may be related to sleep disorders. Exercise intervention can improve sleep quality while changing the diversity of the gut flora, and exercise intervention targeting the gut flora is a new concept for preventing and treating sleep disorders.

## 1. Introduction

Sleep is one of the fundamental physiological processes within the human body. A sufficient quantity and a good quality of sleep are essential requirements for both physical and mental health in humans, as satisfactory sleep restores human physical and mental energy [1]. Recently, inadequate sleep has become a significant global health problem. Disturbing results can be seen in present research on the sleep quality in contemporary students through the Pittsburgh Sleep Quality Index (PSQI) and other research methods—inadequate sleep has become an increasingly significant factor impairing the physical and mental health of students [2,3]. In addition, the intestinal flora is closely related to sleep quality [4]. A study demonstrated that the intestinal flora of insomnia disorder differs significantly in structure and function from that of people with normal sleep patterns [5]. Other research has indicated that physical exercise could replace or supplement existing treatments for sleep problems [6]. As an effective approach to improve sleep, physical exercise can affect the intestinal flora. Research by the American Gut Project showed that partaking in moderate exercise increased the species diversity of *Firmicutes* and promoted a healthier gut environment [7].

The composition of the intestinal microbiota varies with the anatomical part of the intestinal tract, possibly corresponding to the physiological functions of different intestines [8]. With the development of bioinformatics technology, especially 16S rDNA gene sequencing utilized for the analysis of the diversity of the microbiome, a large number of studies have shown that the intestinal microbiota predominantly consists of *Bacteroidetes* and *Firmicutes* at the level of the phylum [9]. For healthy individuals, the intestinal flora lives in symbiosis with us and plays an important role in our physical and mental health. It helps us digest, promotes the absorption of nutrients, produces multiple vitamins and amino acids, neutralizes toxins, and stimulates intestinal transit, especially in the large intestine. Meanwhile, the intestinal flora also performs important roles in the host immune system. Its existence can prevent pathogenic bacteria from rooting in the intestine. It can produce antibacterial substances, kill dangerous bacteria, and stimulate lymphocytes and antibodies [8,9]. At present, considerable research is focused on the relationship between the intestinal microbiome and various diseases such as diabetes, brain disorders, obesity, and cancers [8,10]. Under the studies of brain disorders with the gut microbiota, a concept—the brain–gut axis—is widely accepted in order to explain the gut–brain-microbiome (GBM) interactions [11]. Many recently reported studies have demonstrated that the intestinal microbiota and neurodegenerative diseases (e.g., Alzheimer’s and sleep disorders) affect each other through the brain–gut axis [12].

The authors therefore considered that exercise-induced improvement in sleep quality could be correlated with changes in the intestinal flora. However, most current studies have focused on the link between obesity or overweight and sleep disorders, ignoring studies on sleep quality in low-weight people. There are also great differences in the composition of the intestinal flora of people of different body types, and so we used the low BMI (BMI ≤ 18) student population as the study object. By utilizing an intervention experiment, this study compared the changes in the intestinal flora and the improvement in sleep quality among a group of people with sleep disorders before and after exercise intervention, with the intention of providing additional alternatives in the relief and/or treatment of sleep disorders.

## 2. Materials and Methods

### 2.1. Study Subjects

Here, 671 undergraduates (average age 20.10 ± 1.11) were selected from the Kunming Medical University following screening with the Pittsburgh Sleep Quality Index combined with the results of the National College Students’ Physical Health Test. Before the experiment, all of the subjects were informed of the method, procedure, and purpose of the experiment, and had signed the informed consent form. Subsequently, 11 subjects with sleep disorders (PSQI ≥ 7 and (body mass index) BMI ≤ 18) were selected as the experimental group, and 11 subjects without sleep disorders were selected (PSQI < 7 and BMI ≤ 18) as the control group. These subjects included 4 boys and 18 girls, with an age difference of ±1 year.

### 2.2. Exercise Intervention

The two groups were required to maintain their daily habits and undertake exercise for 8 weeks, in accordance with the specified intervention schedule. One person in each group was chosen to be the team leader to supervise the exercise performance of the team members, in order to ensure that each of them completed the required amount of exercise every week. Each person was required to jog three times per week, at a speed of 8–9 km/h, and for a distance of 4–7 km each time (48–84 km in total over a month) [13]. Both of the subject groups received a weekly telephone or WeChat consultations in which they confirmed that they had maintained their prior dietary and lifestyle regimes.

### 2.3. Stool Collection

Before and after the exercise intervention, the 22 subjects were questioned by utilizing the Pittsburgh Sleep Quality Index, and their stools were collected. The stool samples collected from the experimental group before the commencement of the exercise intervention were defined as “Group E”. The stool samples collected from the control group before the commencement of the exercise intervention were defined as “Group C1”. The stool samples collected from the experimental group after exercise intervention were defined as “Group T”. The stool samples collected from the control group after exercise intervention were defined as “Group C2”. In order to eliminate, as far as possible, the effect of lifestyle habits (e.g., circadian rhythm) on the intestinal flora, each round of stool collection lasted one week, and three stool samples were collected from each subject in each round. Then, three sub-samples (of equal quantities) were taken from the three stool samples and mixed together to form a single sub-sample (the remainder of the samples was frozen in a refrigerator at minus 80 °C).

### 2.4. Extraction and Sequencing of Stool DNA

In this process, 1 g of stool was resuspended in 10 mL of sterile sodium phosphate buffer (0.1 mol/L). The solution was then vortexed for 15 min and centrifuged three times at 200× *g* for 10 min. The coarse particles were discarded and the supernatant was collected. The solution was then centrifuged at 9000× *g* for 10 min to collect the sediment, rinsed four times with 30 mL of sodium phosphate buffer, and finally resuspended in 10 mL of sodium phosphate buffer followed by DNA extraction using the stool DNA extraction kit (QI Aamp DNA Stool MiNiKit). Then, 16S-rDNA Amplicon Sequencing was performed with the V3-V4 region selected, and the conserved region was used to design the universal primers for PCR amplification. The hypervariable region then underwent sequencing and strain identification.

### 2.5. Analysis of Biological Information

In order to study the species diversity in the samples, the effective tags of the collected samples were clustered into OTUs according to a 97% sequence identity, and the representative sequences of the OTUs were then subject to species annotation. According to the results of the OTU clustering, species annotation was added to the representative sequence of each OTU to obtain the corresponding species information and the information about the distribution of species-based abundance. The OTUs were analyzed by abundance, Alpha diversity calculation, and Venn diagrams to obtain the common and unique OTU information for both the exercise group and the control group. The OTUs underwent multi-sequence alignment and the phylogenetic trees were determined. Then, the differences in community structure between the exercise group and the control group were calculated. Finally, principal coordinate analysis (PCoA) was performed.

According to the results of the species annotation of OTUs, the top 10 species for abundance in the phylum, class, order, family, genus, and species in each sample were selected. A bar chart of the relative abundance of these species was generated to visualize the species with a higher relative abundance and their proportions in different categories for each sample. In order to further analyze the differences in community structure between the exercise group and the control group, *t*-test, MetaStat, LEfSe, Anosim, and MRPP were used to analyze the species composition and community structure of the samples. A profiling table was generated to compare the differences between the samples in their abundance of the same species, in order to determine which species had significant differences between the samples. Data diversity analysis was performed to determine the differences in species diversity between the different samples.

### 2.6. Ethical and Data Protection Issues

The study was approved by the Ethic Committee of Kunming Medical University. Participants in this study were volunteers. None of the measurements were known to entail any significant health risk. The study had its own physician to ensure the eligibility and safety of all of the participants. All data were handled and archived confidentially. The benefits and associated risks of the study were carefully explained and the voluntary nature of the participation was emphasized. Informed consent was obtained from all participants prior to the baseline measurements. If the participant agreed to participate, a copy of the signed consent form was kept in their records.

## 3. Results

### 3.1. Demographic Characteristics

Stool samples were collected from 11 patients with sleep disorders and low BMI, as well as from 11 healthy people. Personal data (including age, sex, BMI, smoking, alcohol consumption, and PSQI) were also collected. The demographic characteristics are shown in Table 1. The mean age for the occurrence of sleep disorders was 22.55 ± 1.08. There was no significant demographic difference between the experimental group and the control group. The PSQI score of the experimental group before exercise intervention was significantly higher than that after exercise intervention, and was also higher than the control group (all *p* < 0.001). 

### 3.2. Species Composition Analysis

A community bar chart was generated to analyze the species and their abundance for each group of samples. The results showed that at the level of the phylum, the majority of intestinal flora in the experimental and control groups before and after exercise intervention belonged to *Firmicutes*, *Proteobacteria*, *Bacteroidetes,* and *Actinobacteria*. The dominant phyla in Group E before exercise included *Firmicutes* (70.44%), *Bacteroidetes* (19.79%)*, Proteobacteria* (6.00%), and *Actinobacteriotia* (3.28%), and the top four phyla in Group C1 also included *Firmicutes* (67.50%), *Bacteroidetes* (26.89%)*, Proteobacteria* (3.22%), and *Actinobacteriotia* (2.08%) (Figure 1A). After exercise, the dominant phyla were the same four as before, while the abundance of each in the different groups changed. Both *Firmicutes* and *Actinobacteriotia* in Group T (81.96% and 8.21%, respectively) and Group C2 (71.84% and 4.22%, respectively) increased after exercise; however, the abundance of *Bacteroidetes* and *Proteobacteria* all decreased in the two groups (Group T: 6.99% and 2.59%, Group C2: 20.32% and 3.17%, respectively).

At the level of the genus, bacteria with abundance > 1% were selected for the analysis. The results showed that before exercise, the dominant genera in Group E included *Faecalibacterium*, *Agathobacter*, and *Megamonas*, whose abundances were 13.80%, 10.96%, and 5.77%, respectively; the dominant genera in Group C1 included *Bacteroides*, *Prevotella*, and *Subdoligranulum*, whose abundances were 13.43%, 10.94%, and 4.45%, respectively (Figure 1B). After exercise, *Faecalibacterium* (18.30%), *Blautia* (9.95%), and *Subdoligranulum* (4.45%) became the dominant genera in Group T, while *Agathobacter*, *Bacteroides*, and *Prevotella* decreased in abundance, to 6.95%, 4.08%, and 0.53%, respectively (Figure 1D). In Group C2, *Agathobacter* and *Blautia* (9.95%) increased to 12.91% and 7.69% in abundance, respectively, while *Bacteroides* decreased to 8.48% in abundance (Figure 1F).

### 3.3. Alpha Diversity Analysis

To study the species richness and diversity of the samples, an Alpha diversity analysis was performed on all of the samples. The coverage index of all of the samples was >0.999, indicating a high coverage rate for the samples, in addition to the sequencing results representing the actual situation of the microorganisms in the samples. Indexes of Sobs, Chao, ACE, Shannon, and Simpson were used for the comparison of the alpha diversity between Groups E1 and C1, where the Sobs index referred to the number of OTUs actually observed; the Chao and ACE diversity indexes reflected microbial species richness; and the Shannon and Simpson diversity indexes reflected community richness and evenness. The coverage index was mainly used to measure the sequencing coverage across the species. The difference in the diversity of microbial species between Group E and Group C1 was statistically significant (Table 2); the differences in microbial species richness between Group E and Group T, and between Group C1 and Group C2 were statistically insignificant (Table 3 and Table 4).

### 3.4. Analysis of Beta Diversity

According to the Unifrac distance algorithm, the species diversity among the microbial communities was analyzed between groups using the principal coordinate analysis (PCoA) to analyze the similarities or differences in community composition between the different groups of samples.

During the analysis, the PCoA based on unweighted Unifrac discovered a significant difference (ANOSIM, R = 0.1561, *p* = 0.001) in intestinal flora composition between Groups E and C1, while the PCoA based on weighted Unifrac discovered no difference (ANOSIM, R = 0.0087, *p* = 0.479) (Figure 2A,B). In the comparison of the intestinal flora composition between Groups E and T, both of the results were based on unweighted Unifrac (ANOSIM, R = 0.0895, *p* = 0.021) and based on weighted Unifrac (ANOSIM, R = 0.0892, *p* = 0.044) and indicated a significant difference, which indicated that there was a difference in intestinal flora composition between the patients with sleep disorders before and after exercise intervention (Figure 2C,D), while there was no difference in intestinal flora composition between Groups C1 and C2 (Figure 2E,F).

### 3.5. Analysis of Species Diversity

In order to identify the species with a significant difference between the pre-exercise and post-exercise groups, strict statistical methods were used for hypothesis testing on the species among microbial communities of the groups in order to evaluate the significance level of the difference in species abundance. The results are shown in the figures. At the level of the phylum, the *unclassified_k_norank_d_Bactereia* and *Cyanobacteria* were found to have a significant difference between Groups E and C1 (Figure 3A); *Firmicutes*, *Bacteroidota,* and the *unclassified_k_norank_d_Bacteria* were found to have a significant difference in the change of flora composition between Group E and Group T (Figure 3C), while no difference in the change of flora composition was observed between Group C1 and Group C2 (Figure 3E). At the level of the genus, only *Agathobacter* was found to have a significant difference between Group E and Group C1 (Figure 3B); *Blautia* and *Eubacterium hallii* were found to have a significant difference in the change of flora composition between Group E and Group T (Figure 3D). However, no significant difference in the change of flora was found between Group C1 and Group C2 (Figure 3F).

The LEfSe analysis showed that Group E was abundant in *Negatificutes*, *Veillonellales*-*Selenomonadales*, *Agathobacter*, and *Bacteroidota*; Group C1 was abundant in *Lachnosperaceae_UCG-001*, *Cyanobacteria*, *Eubacterium_xylanophilum* (Figure 4A,B) and *Oxalobacter* (Figure 4E,F); Group T was abundant in *Clostridia* and *Firmicutes* (Figure 4C,D); and Group C2 was abundant in *Lachnospiraceae* (Figure 4E,F).

## 4. Discussion

Currently, more and more college students have sleep problems, and poor sleep quality has become a common issue among college students. A cross-sectional survey about sleep quality among college students in Jilin Province, China, showed that the prevalence of poor sleep quality reached 31%, which is similar to the results of a study on the sleep quality for Taiwan university students [14]. In Chinese and other countries’ studies of sleep quality and BMI, most of the subjects were overweight, and few of these studies focused on the sleep quality of low-BMI people. This study selected people with a low BMI as well as sleep disorders to be research subjects in order to evaluate the sleep quality of college students before and after exercise intervention, as well as details of the change in intestinal flora according to the physiological characteristics of different periods before and after the intervention, with the intention of increasing social awareness regarding the sleep quality of slimmer people.

In this study, the 16srRNA high-throughput sequencing method was utilized to identify the intestinal flora of patients with and without sleep disorders before and after exercise intervention. The analysis of the species composition showed that at the level of the phylum, for the group with sleep disorders after exercise intervention, the abundance of *Firmicutes* and *Actinobacteria* increased, while the abundance of *Bacteroidetes* and *Proteobacteria* decreased; for the healthy group after exercise intervention, the abundance of *Bacteroidetes* decreased. At the level of the genus, it was found that the abundance of *Faecalibactereum*, *Agathobacter*, and *Megamonas* in the group with sleep disorders was higher than that in the healthy group before exercise intervention. In the comparison of before and after exercise intervention, it was found that the abundance of *Faecalibacterium*, *Blautia*, and *Subdoligranulum* in the group with sleep disorders increased, while the abundance of *Agathobacter*, *Bacteroides*, and *Prevotella* decreased; for the healthy people after exercise intervention, the abundance of *Agathobacter* and *Blautia* increased. This indicated that exercise could change the structure and diversity of the intestinal flora. Existing research has found that people who exercise more frequently possess a greater diversity of intestinal flora, which indicates a healthier intestinal environment [15]. It is worth noting that after exercise intervention, the abundance of *Firmicutes* increased to different levels in both the experimental group and the control group, while the abundance of *Bacteroidetes* decreased. This is consistent with the existing research results. Research has shown that the ratio of *Firmicutes* to *Bacteroidetes* (F/B) could be utilized as a potential biological index to evaluate the pathologic status of the human intestinal tract [16]. In the study of the physical exercise levels and intestinal flora, it was also found that the diversity of the phylum *Firmicutes* was the highest, especially for the genus *Faecalibacterium prausnitzii* [17]. In terms of the relationship between sleep quality and microbial species, an increase in *Bacteroidetes* and a decrease in *Firmicutes* were often found in people with insomnia, and studies have shown that *Blautia* and *Ruminococcus,* which belong to the *Firmicutes* phylum, were positively correlated with sleep quality [5,18]. How exactly is the intestinal flora correlated with sleep quality? Several studies have elucidated the possible mechanisms in which some microbes could produce neurotransmitters related to sleep and wakefulness, and influence sleep quality through the brain–gut axis [11,19,20]. However, the current results are not consistent and still need to be further studied.

The alpha diversity analysis showed that there was a statistically significant difference in microbial species richness between the group with sleep disorders and the healthy group (*p* = 0.01), while there was no statistically significant difference in microbial species richness between before and after exercise intervention for both the group with sleep disorders and the healthy group (*p* > 0.05). The beta diversity analysis showed that there was a statistically significant difference in the intestinal flora composition between the group with sleep disorders and the healthy group for the unweighted condition, while no similar result was observed for the weighted conditions; for the healthy group, there was no statistically significant difference in the intestinal flora composition between before and after exercise intervention (*p* > 0.05), while for the group with sleep disorders, there was a statistically significant difference in the intestinal flora composition between before and after exercise intervention (*p* < 0.05). The alpha diversity mainly refers to the species diversity and abundance of a single sample, while the beta diversity refers to the difference in species diversity between groups [21]. Furthermore, the intestinal flora of healthy people will maintain a relatively stable level in adulthood [22,23], and low-intensity exercise may not necessarily lead to changes in the richness and diversity of the intestinal flora (alpha and/or beta diversity) [17]. Therefore, as far as the results of this study were concerned, moderate exercise intervention seemed to have no significant influence on the intestinal flora diversity of the control group students, but the effect was indeed obvious for students with sleep disorders. This is why the composition and diversity of the two groups of intestinal flora have different analysis data before and after the intervention.

The sleep quality scores of the experimental group after the intervention showed that the PSQI scores of the 11 subjects decreased from 8.73 ± 1.66 to 5.00 ± 1.41, which demonstrated that after exercise intervention, the sleep quality of the group with sleep disorders improved and their intestinal flora changed. According to the study on undergraduate men and women, meeting the physical activity guidelines can improve the PSQI scores of college students with sleep issues, which indicates that exercise has a certain curative effect on college students with sleep disorders [24]. Another study showed that the sleep quality of college students was closely correlated with their level of participation in sport activities and that the incidence of sleep disorders among college students undertaking regular sports was significantly lower than among students without regular sports [25]. All of these studies indicated that exercise improved sleep quality.

A rank sum test was conducted to assess the significance level of the difference in species abundance, and the results showed that at the level of the phylum, *Cyanobacteria* differed between the group with sleep disorders and the healthy group, and *Firmicutes* and *Bacteroidota* differed between before and after exercise intervention for the group with sleep disorders. At the level of the genus, the abundance of *Agathobacter* for the group with sleep disorders was higher than that of the healthy group (*p* < 0.05); *Blautia* and *Eubacterium hallii* differed between before and after exercise intervention for the group with sleep disorders, but there was no different flora found among the healthy group.

It was discovered that *Blautia* was correlated with improved sleep quality and that there was a statistically significant increase in the abundance of *Blautia* in the group with sleep disorders who had improved sleep quality after exercise intervention. It has been reported elsewhere that among young healthy people, there is a negative correlation between *Blautia* and *Ruminococcus* and PSQI, and a positive correlation between *Prevotella* and PSQI [26]. *Blautia* has probiotic characteristics, produces high concentrations of acetic acid in vivo to strengthen intestinal epithelial tight junctions, and prevents pathogenic bacteria infection [27], as well as improving prognosis in colon cancer [28] and cirrhosis [29]. A recent study reported that probiotics improved sleep quality and were involved in anti-inflammatory mechanisms [30]. In addition, *Blautia* also participates in the synthesis of short-chain fatty acids (SCFA), a known potent anti-inflammatory substance that is involved in regulatory T cell formation through a mediated immune response, and that is an important source of host and colon epithelium [31].

In addition, it was discovered that *Agathobactor* also affected sleep quality. Several studies have reported that the intestinal microbiota may affect the central nervous system (CNS) by producing SCFA [32]. *Agathobacter* is a type of anaerobic Gram-positive bacterium, and is a new species of *Lachnospiraceae*. The main metabolites of *Agathobacter* are butyric acid, acetic acid, hydrogen, and lactic acid [33]. Butyric acid, a type of SCFA, plays an important role in the intestinal physiology. It has multiple effects on the life cycle of intestinal cells and numerous beneficial effects on health by preventing pathogen invasion, regulating the immune system, and reducing cancer progression [34]. Studies have shown that butyric-acid-producing bacteria and butyric-acid-related metabolites are associated with melatonin [35]. Melatonin is a multi-potent neuroendocrine molecule that is essential to synchronize circadian, sleep/wake cycle, and seasonal rhythms, and has antioxidant, neuroprotective, and immunomodulatory effects [36,37,38]. This indicates that butyric acid is related to sleep quality, and *Agathobacter*, as the main source of butyric acid, can be initially considered as a specific flora that improves sleep quality. However, in this study, no statistically significant changes in *Agathobacter* were observed before or after exercise intervention. This could be because of the heredity, environment, diet, or lifestyle of the subjects, or some other potential interference factors could have influenced the experimental results.

*Eubacterium hallii* was also a significant discovery in this study: its abundance changes also affected sleep quality. Udayappan et al. observed that after the administration of *Eubacterium hallii*, obese and diabetic db/db mice were discovered to have improved insulin sensitivity and energy metabolism. This effect resulted from the ability of *Eubacterium hallii* to produce SCFA when fermenting glucose and intermediates [39]. Other researchers have indicated that this metabolic versatility is an important factor in the ability of *Eubacterium hallii* to restore the homeostasis of host intestinal microbiota and to improve the colitis induced by dextran sulfate sodium (DSS) in mice [40]. This supported the specific interaction between the intestinal flora and sleep quality. The effect of intestinal flora on sleep quality could be based on its metabolic products and pathways and inflammatory response. Research has shown that sleep and circadian rhythms may disrupt the composition of the microbiota through inflammation and the destruction of epithelial barriers [41]. Inflammatory response mechanisms are induced by the microbiota, and then trigger the central nervous system, thereby exacerbating insomnia and depression [42]. Accordingly, this study confirmed the correlation between the intestinal flora, the potential therapeutic effects of the intestinal flora regulation for sleep improvement, and the indispensable role of exercise as one of the methods for regulating the intestinal flora.

There are also some limitations associated with these results. Firstly, because of the limited conditions for the experiment, in this study, only 22 college students were selected as experimental subjects, a sample size too small to ascertain additional relevant factors. It is recommended that follow-up studies should increase the sample size and expand the subject groups. Secondly, the quality control of the samples in this study needed to be enhanced: the histories of medication use of the research subjects should have been collected, their eating habits should have been controlled as far as possible to avoid significant variations, and the time of sampling and the delivery of samples should have been controlled to improve the reliability of the results. In addition, because 16S-rDNA amplification sequencing was adopted as a result of cost considerations, many of the sequences obtained by 16S-rDNA sequencing were not annotated at the level of species, causing certain limitations in identifying specific species. However, macrogenome sequencing can randomly break the genomic DNA of a microorganism into small fragments and then splice them into long sequences by the assembly. Therefore, macrogenome sequencing is highly advantageous in species identification. Finally, with the widespread application of genomics, transcriptionology, proteomics, and metabonomics, the prediction of the functions of specific strains, immune status, intestinal barrier integrity, and related information about metabolomics should be included in future studies to provide additional options for clinical diagnosis and treatment.

## 5. Conclusions

In conclusion, there is a difference in the abundance of intestinal flora between people with sleep disorders and healthy people. Exercise intervention not only changes the diversity of the intestinal flora of people with sleep disorders, but is also an important means for improving sleep quality. In this study, it was discovered that the abundance of *Blautia* and *Eubacterium hallii* increased, while *Agathobactor* decreased after exercise intervention, and the composition of the gut flora changes were conducive to an improvement in sleep quality, indicating that the intestinal flora may be related to sleep quality. This study preliminarily explored the relationship between exercise intervention, intestinal flora, and sleep quality, and suggests that exercise intervention may help improve sleep quality by promoting changes in the structure of the intestinal flora, but the cause–effect relationship and mechanisms of the three still need to be studied further.

## Figures and Tables

**Figure 1 ijerph-19-12385-f001:**
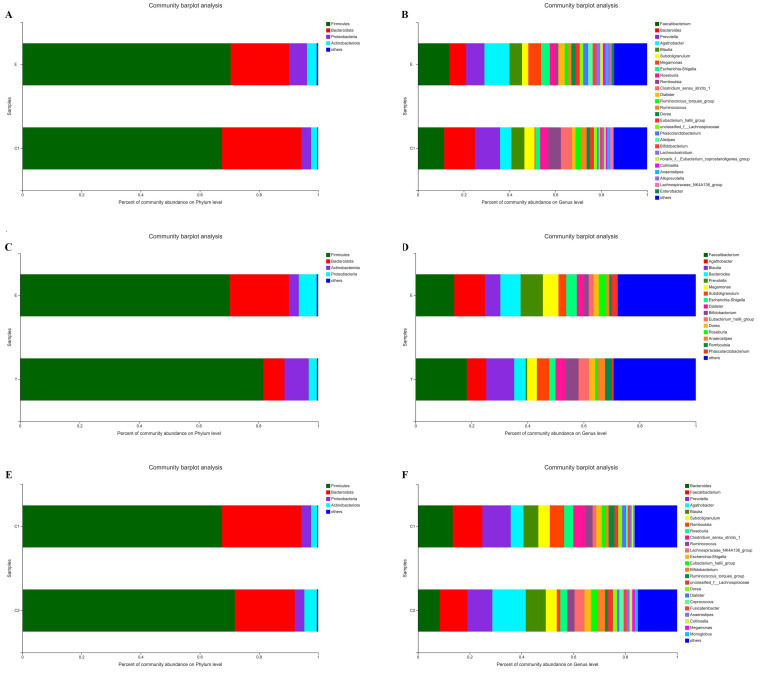
Comparisons of the intestinal microbiome composition in different groups before and after exercise intervention. (**A**,**C**,**E**) The relative abundance of each phylum in each group. (**B**,**D**,**F**) The relative abundance of each genus in each group. Different colors represent different phyla or genera. E: the experimental group before exercise intervention; T: the experimental group after exercise intervention; C1: the control group before exercise intervention; C2: the control group after exercise intervention.

**Figure 2 ijerph-19-12385-f002:**
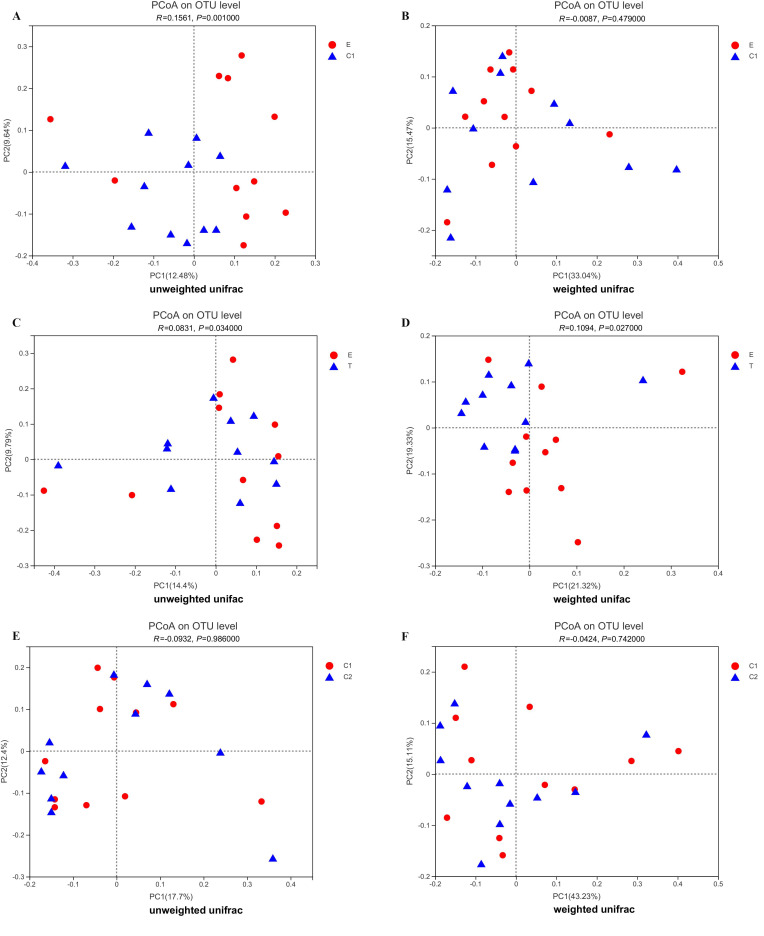
PCoA based on the unweighted UniFrac metrics (**A**,**C**,**E**) and weighted UniFrac metrics (**B**,**D**,**F**) of the fecal samples from the pre−exercise and post−exercise groups. Each shape in the figure represents a sample, and shapes of different colors indicate different groups. The percentage in the coordinate axis bracket represents the proportion of sample difference data (distance matrix), which the corresponding coordinate axis can explain. PCoA: principal coordinate analysis; E: experimental group before exercise intervention; T: experimental group after exercise intervention; C1: control group before exercise intervention; C2: control group after exercise intervention.

**Figure 3 ijerph-19-12385-f003:**
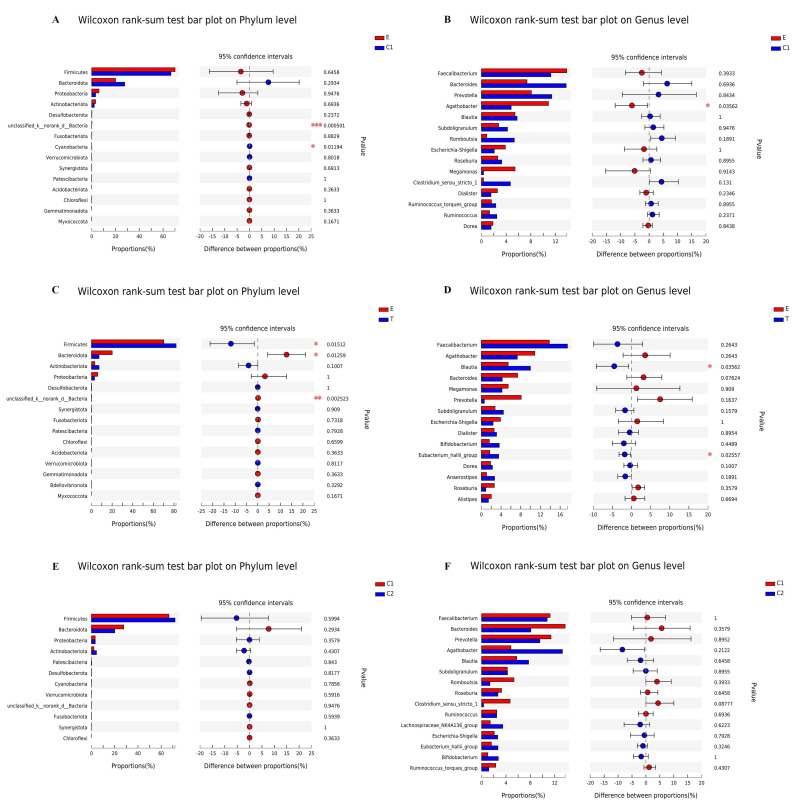
Evaluating the significance level of the difference in species abundance at the level of phylum or genus between groups before and after exercise intervention. (**A**,**C**,**E**) The phylum level to compare the difference in gut microbiota between groups. (**B**,**D**,**F**) The genus level to compare the difference in gut microbiota between groups. Different colors represent different groups. E: experimental group before exercise intervention; T: experimental group after exercise intervention; C1: control group before exercise intervention; C2: control group after exercise intervention. * *p* < 0.05, ** *p* < 0.01, *** *p* < 0.001.

**Figure 4 ijerph-19-12385-f004:**
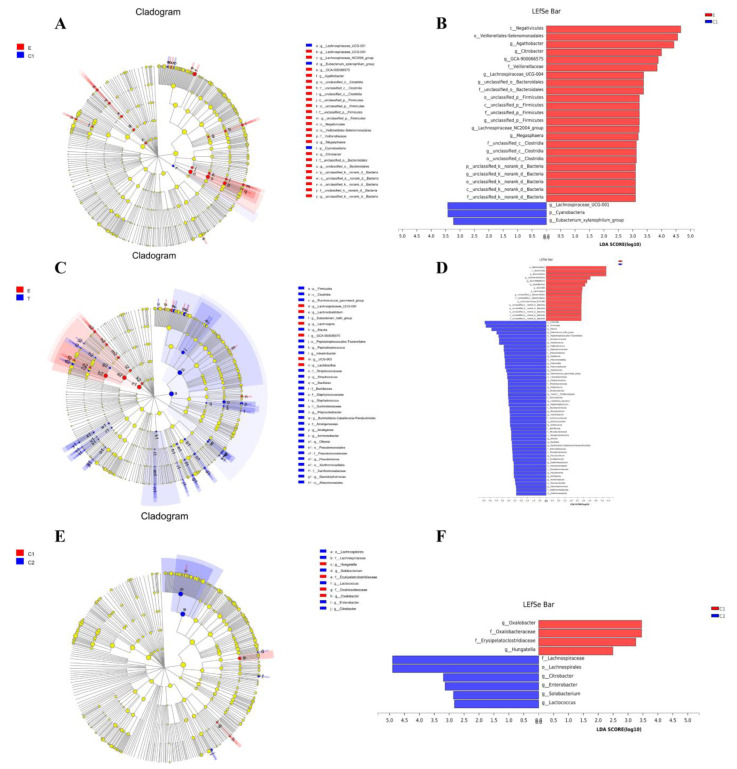
LEfSe analysis of the difference in the abundance of the gut microbiota in each group. (**A**,**C**,**E**) Cladograms showing the taxonomic hierarchy of the main taxa from the phylum to genus (from inner to outer) in the sample community. (**B**,**D**,**F**) LDA score showing the abundance of different taxa in the intestinal flora between the groups. LEfSe: linear discriminant analysis effect size; LDA: linear discriminant analysis; E: experimental group before exercise intervention; T: experimental group after exercise intervention; C1: control group before exercise intervention; C2: control group after exercise intervention.

**Table 1 ijerph-19-12385-t001:** Demographic characteristics of the experimental group and the control group.

Parameter	Experimental Group (*n* = 11)	Control Group (*n* = 11)	*x^2^*/*t*/*z*	*p* Value
Age, Years	22.55 ± 1.08	21.82 ± 1.27	1.384	0.182
Gender, female/male	7/4	11/0	—	0.090
BMI	17.65 ± 0.90	17.73 ± 1.19	−0.156	0.878
Smoking, yes/no	2/9	0/11	—	0.090
Drinking, yes/no	2/9	1/10	—	0.090
PSQI				
pre-exercise	8.73 ± 1.66	4.64 ± 0.88	6.895	<0.001
post-exercise	5.00 ± 1.41 *	—	—	—

Note: BMI, body mass index; PSQI, Pittsburgh Sleep Quality Index. * *p* < 0.001 compared with pre-exercise.

**Table 2 ijerph-19-12385-t002:** Indexes of the intestinal flora diversity of the pre-exercise sleep—disordered Group (E) and sleep—normal Group (C1).

Parameter	E	C1	*p* Value
Sobs	695.73 ± 130.22	437.09 ± 81.02	<0.001
Shannon	3.85 ± 0.44	3.63 ± 0.51	0.27
Simpson	0.07 ± 0.04	0.07 ± 0.04	0.78
Ace	760.25 ± 146.67	588.35 ± 103.20	0.01
Chao	715.75 ± 134.95	569.54 ± 102.44	0.01
Coverage	1.00 ± 0.00	1.00 ± 0.00	<0.001

**Table 3 ijerph-19-12385-t003:** Indexes of the intestinal flora diversity of the sleep—disordered group before and after exercise intervention.

Parameter	E	T	*p* Value
Sobs	695.73 ± 130.22	512.55 ± 88.59	<0.001
Shannon	3.85 ± 0.44	3.90 ± 0.23	0.77
Simpson	0.07 ± 0.04	0.05 ± 0.01	0.26
Ace	760.25 ± 146.67	691.56 ± 146.48	0.29
Chao	715.75 ± 134.95	673.28 ± 126.53	0.46
Coverage	1.00 ± 0.00	1.00 ± 0.00	<0.001

Note: E means pre-exercise; T means post-exercise.

**Table 4 ijerph-19-12385-t004:** Indexes of intestinal flora diversity of the sleep—normal group before and after exercise intervention.

Parameter	C1	C2	*p* Value
Sobs	437.09 ± 81.02	411.64 ± 95.90	0.51
Shannon	3.63 ± 0.51	3.61 ± 0.37	0.95
Simpson	0.07 ± 0.04	0.07 ± 0.03	0.80
Ace	588.35 ± 103.20	563.19 ± 154.67	0.66
Chao	569.54 ± 102.44	551.30 ± 141.67	0.73
Coverage	1.00 ± 0.00	1.00 ± 0.00	0.11

Note: C1 means pre-exercise; C2 means post-exercise.

## Data Availability

The data that support the findings of this study are available upon request from the corresponding author.

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
