# Peer review of "Exercise Interventions Improved Sleep Quality through Regulating Intestinal Microbiota Composition"

_ijerph, 2022, doi:10.3390/ijerph191912385_

Round 1

Reviewer 1 Report

The authors used sleep disordered group vs. healthy groups, compared before and after exercise intervention effect, revealed that exercise could change the diversity of gut flora thus improving sleep quality. Most parts of this manuscript show that exercise changes intestinal flora diversity, but lack of the evidence that sleep quality is improved after exercise.

In table 1 and line 318, could you add the information of PSQI in T and C2 groups showing that sleep quality is improved after exercise in C2?

In figure 1, Firmicutes is higher in E compared to C1, and after exercise Firmicutes increases in both T and C2, and Bacteroidetes shows the opposite pattern. It is quite interesting, could you discuss more about them related to sleep quality.

In figure 1, the colors representing species are not consistent, e.g. Figure 1A purple indicates Proteobacteria, while in Figure 1C purple indicates Actinobacteriota.

 In table 2, 3, 4 and Figure 2, alpha diversity analysis shows no significant difference in E and T, C1 and C2; while beta diversity analysis shows significant difference in E and T, but still no significant difference in C1 and C2, are there some explanations for these differences?

At last, I think Figure 3 include all the content of Figure 1, and Figure 3 contains significant test, Figure 1 is kind of redundant.

Author Response

We appreciate your careful reminding. Your comments are very valuable and helpful for revising our paper. All changes had been made with red color in the revised manuscript

Point 1: The authors used sleep disordered group vs. healthy groups, compared before and after exercise intervention effect, revealed that exercise could change the diversity of gut flora thus improving sleep quality. Most parts of this manuscript show that exercise changes intestinal flora diversity, but lack of the evidence that sleep quality is improved after exercise. In table 1 and line 318, could you add the information of PSQI in T and C2 groups showing that sleep quality is improved after exercise in C2?

Response 1: Thank you for your comment. Your suggestion is very valuable and helpful. We have added the information of PSQI of the experimental group after exercise intervention in Table 1 and line 318. And we changed "T" and "C1" to "Experimental group" and "Control group" respectively to make the expression more accurate. Meanwhile, because there were no sleep disorders in the control group, we did not perform PSQI on the control group after the exercise intervention. Table 1 is modified as follows.

“3.1 Demographic characteristics: There was no significant demographic difference between the experimental group and the control group. The PSQI score of the experimental group before exercise intervention was significantly higher than that after exercise intervention, also higher than the control group (all P < 0.001).

Table 1 Demographic Characteristics of the Experimental Group and the Control Group

Parameter

Experimental group (N=11)

Control group (N=11)

x2/t/z

P value

Age, Years

22.55±1.08

21.82±1.27

1.384

0.182

Gender, female/male

7 / 4

11 / 0

0.090

BMI

17.65±0.90

17.73±1.19

-0.156

0.878

Smoking, yes/no

2 / 9

0 / 11

0.090

Drinking, yes/no

2 / 9

1 / 10

0.090

PSQI

  pre-exercise

8.73±1.66

4.64±0.88

6.895

<0.001

  post-exercise

5.00±1.41*

Note: BMI, body mass index; PSQI, Pittsburgh Sleep Quality Index. * P < 0.001 compared with the pre-exercise.”

Point 2: In figure 1, Firmicutes is higher in E compared to C1, and after exercise Firmicutes increases in both T and C2, and Bacteroidetes shows the opposite pattern. It is quite interesting, could you discuss more about them related to sleep quality.

Response 2: Thank you for your advice. We add the following information in the Discussion.

“And in the study of physical exercise levels and intestinal flora, it was also found that the diversity of phylum Firmicutes was the highest, especially genus Faecalibacterium prausnitzii [17]. In terms of the relationship between sleep quality and microbial, an increase in Bacteroidetes and a decrease in Firmicutes were often found in people with insomnia, and studies showed that Blautia and Ruminococcus, which belong to the Firmicutes phylum, were positively correlated with sleep quality [18,19]. And how exactly the intestinal flora correlates with sleep quality? Several studies had elucidated possible mechanisms in that some microbes could produce neurotransmitters related to sleep and wakefulness and influence sleep quality through the brain-gut axis [20-22]. However, the current results are not consistent and still need to be further studied."

Point 3: In figure 1, the colors representing species are not consistent, e.g. Figure 1A purple indicates Proteobacteria, while in Figure 1C purple indicates Actinobacteriota.

Response 3: Thank you for your reminder. We carefully checked all the pictures in this manuscript and found two places that needed to be corrected. They have all been corrected as follows: (1) The colors representing Proteobacteria and Actinobacteriota in Figure 1C had been modified to be consistent with Figures 1A and 1E (Figure 1). (2) Legend “E1” in Figures 2A and 2B were all changed to “E” (Figure 2).

Point 4: In table 2, 3, 4 and Figure 2, alpha diversity analysis shows no significant difference in E and T, C1 and C2; while beta diversity analysis shows significant difference in E and T, but still no significant difference in C1 and C2, are there some explanations for these differences?

Response 4: Thank you for your advice. We add the following information in the Discussion.

“And alpha diversity mainly refers to the species diversity and abundance of a single sample, while beta diversity refers to the difference in species diversity between groups [23]. Furthermore, the intestinal flora of healthy people will maintain a relatively stable level in adulthood [24,25], and low-intensity exercise may not necessarily lead to changes in the richness and diversity of the intestinal flora (alpha and/or beta diversity) [17]. Therefore, as far as the results of this study were concerned, that moderate exercise intervention might have no significant influence on the intestinal flora diversity of the control group students, but the effect is indeed obvious for students with sleep disorders. This is why the composition and diversity of the two groups of intestinal flora will have different analysis data before and after the intervention.”

Point 5: At last, I think Figure 3 include all the content of Figure 1, and Figure 3 contains significant test, Figure 1 is kind of redundant.

Response 5: Thank you for your advice. We think that the stack plot in Figure 1 is intended to provide a more visual representation of the species composition between groups, while Figure 3 is primarily a hypothesis test of the abundance of each species between two groups to see if there is a statistically significant difference. So we think that both of the two pictures are necessary in this manuscript.

Reviewer 2 Report

Abstract and  Results- Analysis of species diversity

·         name group/phylum must not contain "_".

Ex.  Eubacterium_hallii

·         “Blautia, Eubacterium_hallii_group, and Agathobacter are associated with healthy sleep.”

Please mention if the increase or decrease of these bacteria is associated with healthy sleep.

Manuscript

Introduction

The introduction needs to be improved. It is very short. I recommend that you add some information about the composition of the gut microbiota, its role in the body's physiological function and in maintaining the body's homeostasis. You can include a phrase about the brain-gut axis.

Materials and Methods

2.1 Study subjects

·         Why did you choose subjects with low BMI (below 18)? You have the explanation in the discussions, but it's important to mention here as well.

·         You specify the number of girls and boys in both groups, but it is important to state this for each group (for the control group and for the experimental group).

2.2 Exercise intervention

“Two groups of subjects were selected for this study, and they were divided into the 70 experimental group and the control group.” You have this information in the study subjects.

Results

Lines 158-161: You specify three dominated phyla with their percentage for group E, but only Firmicutes for group C1. Can you explain why or please fill in the percentage for Proteobacteria and Actinobacteriotia  in group C1 as well.

I recommend a legend for each figure 1, 2, 3 (1A, IB, 2A, etc.). The legends of figure 1,2 and 3 contain a lot of information that are too difficult to read.

Conclusion

Line 408-409: “indicating that intestinal flora could be associated with both sleep and disease severity.” Which disease? Insomnia? Did you assess the severity of the subjects' sleep disturbance?

I recommend presenting the conclusions of the study in a more concise manner.

Minor comments

·         Please insert a space between words and references

Ex. students[2,3]      students [2,3]

Author Response

We appreciate your careful reminding. Your comments are very valuable and helpful for revising our paper. All changes had been made with red colored text in the current manuscript and our itemized responses to your comments, questions, and suggestions (repeated below for your convenience) are as follows.

Point 1: Comments and Suggestions for Authors: Abstract and Results-Analysis of species diversity

  • name group/phylum must not contain "_". Ex. Eubacterium_hallii
  • Blautia, Eubacterium_hallii_group, and Agathobacter are associated with healthy sleep.”

Please mention if the increase or decrease of these bacteria is associated with healthy sleep.

Response 1: Thank you for your advice. We have checked full text and modified all similar “Eubacterium_hallii_group” cases. And the corresponding place in the article has indicated whether these bacteria related to healthy sleep are increased or decreased.

Point 2: Introduction: The introduction needs to be improved. It is very short. I recommend that you add some information about the composition of the gut microbiota, its role in the body's physiological function and in maintaining the body's homeostasis. You can include a phrase about the brain-gut axis.

Response 2: Thank you for your advice. Your suggestion is very valuable and helpful. We have add the information about the composition of the gut microbiota and its function in the host, and brought up the concept of the brain-gut axis. At the same time, we have added a detailed description in the main text with appropriate references, the following parts marked in red are the added sentences in Introduction.

“The composition of the intestinal microbiota varies with the anatomical part of the intestinal tract, maybe corresponding to the physiological functions of different intestines [8]. With the development of bioinformatics technology, especially 16S rDNA gene sequencing utilized for the analysis of the diversity of microbiome, a large number of studies have shown that the human intestinal microbiota predominantly consists of Bacteroidetes and Firmicutes at the level of phylum [9]. For healthy individuals, the intestinal flora lives in symbiosis with us and plays an important role in our physical and mental health. It helps us digest, promotes the absorption of nutrients, produces multiple vitamins and amino acids, neutralizes toxins, and stimulates intestinal transit, especially in the large intestine. Meanwhile, the intestinal flora also performs important roles in the host immune system. Its existence can competitively prevent pathogenic bacteria from rooting in the intestine. It can produce antibacterial substances, kill dangerous bacteria, and stimulate lymphocytes and antibodies [8,9]. At present considerable research focus on the relationship between intestinal microbiome and various diseases such as diabetes, brain disorders, obesity, and cancers [8,10]. Under the studies of brain disorders with gut microbiota, a conception–the brain-gut axis–is widely accepted to explain the gut-brain-microbiome (GBM) interactions [11]. And many recently reported studies to demonstrate that intestinal microbiota and neurodegenerative diseases (e.g., Alzheimer’s, sleep disorder) affect each other through the brain-gut axis [12].”

Point 3: 2.1 Study subjects

  • Why did you choose subjects with low BMI (below 18)? You have the explanation in the discussions, but it's important to mention here as well.
  • You specify the number of girls and boys in both groups, but it is important to state this for each group (for the control group and for the experimental group).

Response 3: Thank you for your advice. (1) We added the reasons for choosing low BMI (below 18) people for the study in the Introduction. The additions are indicated in red as follows.

“However, most current studies have focused on the link between obesity or overweight and sleep disorders, ignoring studies on sleep quality in low-weight people. And there are also great differences in the composition of the intestinal flora of people of different body types, so we took the low BMI (BMI ≤ 18) student population as the study object.”

(2) In this manuscript, the number of girls and boys in each group is already mentioned in Table 1, so this information is not repeated in the Study subjects.

Point 4: 2.2 Exercise intervention:“Two groups of subjects were selected for this study, and they were divided into the 70 experimental group and the control group.” You have this information in the study subjects.

Response 4: Thank you for your reminder. This information is indeed mentioned in the Study subjects, we have deleted this sentence in this manuscript.

Point 5: Results: Lines 158-161: You specify three dominated phyla with their percentage for group E, but only Firmicutes for group C1. Can you explain why or please fill in the percentage for Proteobacteria and Actinobacteriotia in group C1 as well.

Response 5: Thank you for your reminder. We have filled in the percentage for Proteobacteria and Actinobacteriotia in group C1, and readjusted the formulation of the corresponding sentences. The following parts marked in red are readjusted.

“The dominant phyla in group E before exercise included Firmicutes (70.44%), Bacteroidetes (19.79%), Proteobacteria (6.00%) and Actinobacteriotia (3.28%), and the top four phyla in group C1 also included Firmicutes (67.50%), Bacteroidetes (26.89%), Proteobacteria (3.22%) and Actinobacteriotia (2.08%) (Figure 1A). After exercise, the dominant phyla were still the four while the abundance of each in different groups changed. Both Firmicutes and Actinobacteriotia in group T (81.96% and 8.21%, respectively) and group C2 (71.84% and 4.22%, respectively) were increased after exercise; however, the abundance of Bacteroidetes and Proteobacteria were all decreased in the two groups (group T: 6.99% and 2.59%, group C2: 20.32% and 3.17%, respectively).”

Point 6: Results: I recommend a legend for each figure 1, 2, 3 (1A, IB, 2A, etc.). The legends of figure 1,2 and 3 contain a lot of information that are too difficult to read.

Response 6: Thank you for your advice. Your suggestion is very valuable and helpful. We have revised the legends of the four pictures in this manuscript, and the following parts are the rewritten legends:

FIGURE 1 Comparisons of intestinal microbiome composition in different groups before or after exercise intervention. (A, C, E) The relative abundance of each phylum in each group. (B, D, F) The relative abundance of each genus in each group. Different colors represent different phyla or genera. E: the experimental group before exercise intervention; T: the experimental group after exercise intervention; C1: the control group before exercise intervention; C2: the control group after exercise intervention.

FIGURE 2 PCoA based on the unweighted UniFrac metrics (A, C, E) and weighted UniFrac metrics (B, D, F) of fecal samples from the pre-exercise and post-exercise groups. Each shape in the figure represents a sample, and shapes of different colors indicate different groups. The percentage in the coordinate axis bracket represents the proportion of sample difference data (distance matrix) that the corresponding coordinate axis can explain. PCoA: Principal coordinate analysis; E: the experimental group before exercise intervention; T: the experimental group after exercise intervention; C1: the control group before exercise intervention; C2: the control group after exercise intervention.

FIGURE 3 Evaluating the significance level of the difference in species abundance at the level of phylum or genus between Groups before and after exercise intervention. (A, C, E) were at the phylum level to compare the difference in gut microbiota between groups. (B, D, F) were at the genus level to compare the difference in gut microbiota between groups. Different colors represent different groups. E: the experimental group before exercise intervention; T: the experimental group after exercise intervention; C1: the control group before exercise intervention; C2: the control group after exercise intervention.

FIGURE 4 LEfSe analysis of the difference in the abundance of gut microbiota in each group. (A, C, E) were cladograms showing the taxonomic hierarchy of the main taxa from phylum to genus (from inner to outer) in the sample community. (B, D, F) were LDA score showing the abundance of different taxa in intestinal flora between the groups. LEfSe: Linear discriminant analysis effect size; LDA: linear discriminant analysis; E: the experimental group before exercise intervention; T: the experimental group after exercise intervention; C1: the control group before exercise intervention; C2: the control group after exercise intervention.”

Point 7: Conclusion: Line 408-409: “indicating that intestinal flora could be associated with both sleep and disease severity.” Which disease? Insomnia? Did you assess the severity of the subjects' sleep disturbance? I recommend presenting the conclusions of the study in a more concise manner.

Response 7: Thank you for your advice. Based on the research data in this study, we have recalibrated the formulation and further summarized the points of the article. The following red part is the revised conclusions.

“In this study, it was discovered that the abundance of Blautia, Eubacterium hallii group increased while Agathobactor decreased after exercise intervention, and the composition of gut flora changes are conducive to the improvement of sleep quality, indicating that intestinal flora may be related to sleep quality. This study preliminarily explores the relationship between exercise intervention, intestinal flora and sleep quality, and suggests that exercise intervention may help improve sleep quality by promoting changes in the structure of intestinal flora, but the cause-effect relationship and mechanisms of the three still need to be further studied.”

Point 8: Minor comments: Please insert a space between words and references. Ex. students[2,3]      students [2,3]

Response 8: Thank you for your advice. We have checked the full text and inserted a space between all references and words.

Round 2

Reviewer 1 Report

I think the author answered my questions well.